

# Metabolic and genomic analysis elucidates strain-level variation in *Microbacterium spp.* isolated from chromate contaminated sediment

Michael W. Henson[1], Jorge W. Santo Domingo[2], Peter S. Kourtev[3], Roderick V. Jensen[4], James A. Dunn[1] and Deric R. Learman[1]

[1] Institute for Great Lakes Research and Department of Biology, Central Michigan University, Mount Pleasant, MI, United States
[2] National Risk Management Research Laboratory, Environmental Protection Agency, Cincinnati, OH, USA
[3] Department of Biology, Central Michigan University, Mount Pleasant, MI, United States
[4] Department of Biological Sciences, Virginia Polytechnic Institute and State University (Virginia Tech), Blacksburg, VA, United States

## ABSTRACT

Hexavalent chromium [Cr(VI)] is a soluble carcinogen that has caused widespread contamination of soil and water in many industrial nations. Bacteria have the potential to aid remediation as certain strains can catalyze the reduction of Cr(VI) to insoluble and less toxic Cr(III). Here, we examine Cr(VI) reducing *Microbacterium* spp. (Cr-K1W, Cr-K20, Cr-K29, and Cr-K32) isolated from contaminated sediment (Seymore, Indiana) and show varying chromate responses despite the isolates' phylogenetic similarity (i.e., identical 16S rRNA gene sequences). Detailed analysis identified differences based on genomic metabolic potential, growth and general metabolic capabilities, and capacity to resist and reduce Cr(VI). Taken together, the discrepancies between the isolates demonstrate the complexity inter-strain variation can have on microbial physiology and related biogeochemical processes.

## INTRODUCTION

Chromium (Cr) has become a major environmental pollutant due to its extensive use in industrial and mining industries (*Barak et al., 2006*; *Brose & James, 2010*; *Cheng, Holman & Lin, 2012*). Chromium is the second most common contaminant at US Department of Energy sites, and without proper remediation could persist at dangerous levels well past 1,000 years from now (*Okrent & Xing, 1993*; *Riley, Zachara & Wobber, 1992*). Further, chromium is of particular concern because of its solubility in water and potential risk to drinking water; as well as the potential to inhibit the natural microbial communities engaged in the bioremediation of other pollutants (*Bååth, 1989*; *Gough et al., 2008*; *Gough & Stahl, 2011*; *Kourtev, Nakatsu & Konopka, 2009*; *Nakatsu et al., 2005*; *Said & Lewis, 1991*). As such, chromium contamination of soil and water poses risks to the United States and

Corresponding author
Deric R. Learman,
deric.learman@cmich.edu

other industrial nations without proper remediation (*Ackerley et al., 2004a*; *Beller et al., 2013*; *Cheng, Holman & Lin, 2012*; *He et al., 2010*).

Within the environment, chromium mainly persists in two forms: Cr(III) and Cr(VI) (*Bartlett, 1991*). Cr(VI) is highly toxic, soluble, and can be easily transported across cell membranes of both eukaryotic and prokaryotic organisms via sulfate and other active transporters (*Ackerley et al., 2004b*; *Cheng, Holman & Lin, 2012*). Conversely, Cr(III) is sparingly soluble, thermodynamically stable, and found in oxide, hydroxide, and sulfate minerals, or complexed by organic matter (*Barak et al., 2006*; *Brose & James, 2010*; *Oze, Bird & Fendorf, 2007*; *Oze et al., 2004*; *Viti et al., 2013*). While the natural oxidation of Cr(III) is only mediated by a select number of abiotic substrates (e.g., manganese oxides and hydrogen peroxide), Cr(VI) reduction can be driven by various bacterial and abiotic factors (e.g., iron(II) and hydrogen sulfides) (*Brose & James, 2010*; *Oze et al., 2004*; *Viti et al., 2013*). Chromium (VI) compounds are highly toxic, mutagenic, and carcinogenic. Many microorganisms have been found to possess various mechanisms to reduce Cr(VI), which has potential impacts on bioremediation strategies (*Suzuki et al., 1992*).

Several bacterial species have been isolated and characterized detailing their unique mechanisms to reduce and resist Cr(VI) (*Chovanec et al., 2012*; *Diaz-Perez et al., 2007*; *Field et al., 2013*; *He et al., 2010*; *Silver & Phung, 2005*). While certain bacteria can reduce Cr(VI) under anaerobic conditions (*Beller et al., 2013*; *Chovanec et al., 2012*; *Lovley et al., 1993*; *Sheik et al., 2012*), other bacteria possess soluble enzymes that facilitate reduction under aerobic conditions (*Ackerley et al., 2004a*; *Barak et al., 2006*; *Cheung & Gu, 2007*; *Gonzalez et al., 2005*; *Park et al., 2000*). In addition, bacteria can also utilize an efflux pump to aid resistance to chromate. ChrA, part of the CHR superfamily, is a chromium transport protein that has been linked to Cr(VI) resistance. The mechanism of this efflux pump has been determined by examining two model organisms, *Cupravidus metallidurans* and *Pseudomonas aeruginos* (*Cervantes & Ohtake, 1988*; *Cervantes et al., 1990*). In an *in silico* study, *Ramirez-Diaz et al. (2008)* identified 135 ChrA orthologs that were dominated by *Proteobacteria* representatives.

Previous studies have shown that chromate reduction under aerobic conditions can be catalyzed by soluble NADH/NADPH dependent oxidoreductases. Two of these chromate reductases have been well studied: ChrR in *Pseudomonas putida* (*Park et al., 2000*) and YieF in *Escherichia coli* (*Barak et al., 2006*). ChrR uses a one electron transfer to reduce Cr(VI) to Cr(V), a reactive intermediate, and then a second electron transfer to generate Cr(III). While the reactive intermediate can re-oxidize into Cr(VI) (in the presence of oxygen), ChrR uses quinone reductase activity to help protect cells against reactive oxygen species (ROS) production (*Ackerley et al., 2004a*; *Cheung & Gu, 2007*; *Gonzalez et al., 2005*). YieF, a sequence homolog of ChrR, utilizes a four electron transfer to reduce Cr(VI) to Cr(III). The reductive mechanism of YieF also produces ROS, similar to ChrR, however, at a much lower rate (*Ackerley et al., 2004b*; *Ramirez-Diaz et al., 2008*).

Members of the genus *Microbacterium* have been shown to reduce chromate, however, the genes involved are not fully resolved. Indeed this is the case for *Microbacterium sp.* SUCR140 (*Soni et al., 2014*), *Microbacterium sp.* chr-3 (*Focardi, Pepi & Focardi, 2013*), and

*Microbacterium sp.* CR-07 (*Liu et al., 2012*), as each of the isolates have been shown to be effective reducers of chromate. Another example is the case for *M. laevaniformans* strain OR221 (*Brown et al., 2012*), a metal resistant bacterium whose genome has been described, however, its genome does not contain an annotated *chrR* or *yieF*. Here, we conducted a combined genomic, metabolic, and physiological analysis of chromate reducing bacteria from the *Microbacterium* genus. Though the isolates were identical at the 16S rRNA level, we found considerable inter-strain genomic, metabolic, and Cr(VI) reduction and resistance variability. This, taken together with their ability to resist and reduce high amounts of chromate, makes these isolates an interesting case study on microdiversity within highly related taxa as well as on chromate reduction.

## METHODS & MATERIALS

### Sample collection and isolation

Soil was collected in Seymour, IN from a Department of Transportation site known to have contamination from chromium, lead, and organic solvents (*Kourtev, Nakatsu & Konopka, 2006*; *Nakatsu et al., 2005*). Bill Jervis from the Indiana Department of Transport provided site access (no permit was required) and the project did not involve endangered or protected species. Bacterial strains were isolated from the contaminated soil as described in *Kourtev, Nakatsu & Konopka (2009)*. Briefly, isolates were enriched on 50% trypic soy agar (TSA) and 0.25 mM Cr(VI). Colonies were picked from the TSA plates and placed in 50% TSB with .25 mM Cr(VI). Individual isolates were grown at varying amounts of chromate enriched TSA and tryptic soy broth (TSB). The isolates were allowed to grow and then selected based on their varying abilities to resist and reduce Cr(VI). Isolates were maintained on 50% TSA with 2 mM Cr(VI) as well as in 30% glycerol stocks stored at $-80\,°C$ to help preserve active pure cultures.

### DNA extraction

Bacterial isolates used in this study (Cr-K1W, Cr-K20, Cr-K29, and Cr-K32) were grown in 250 ml of 50% TSB amended with 2 mM potassium chromate ($K_2CrO_4$). Following inoculation, isolates were incubated at $30\,°C$, 225 rpm for 24–72 h. Cells were harvested by centrifuging the cultures in 250 ml Nalgene bottles at $8,000 \times g$ for 18 min. The supernatant was removed and the pellet was washed with autoclaved nanopure water following the same procedure. Cell pellets were extracted using the FastDNA® Spin Kit (MP Biomedical, Santa Ana, CA) with one modification. Before extraction, cell pellets were resuspended in autoclaved nanopure water before 200 µl was transferred to the Lysis Matrix tube provided by the kit. DNA samples were recovered in 100 µl of DES solution and stored at $-20\,°C$.

### Chromate resistance and reduction experiments

All growth and reduction experiments were conducted in 50% TSB at $30\,°C$ while shaking at 225 rpm. Chromium was added when needed in the form of $K_2CrO_4$. Growth was measured on a UV-Vis spectrophotometer (Varian Cary 50, Agilent) at 600 nm. To test the ability of isolates to reduce Cr(VI), chromate was added to the media to a final

concentration of 2 mM. Cultures, inoculated from $-80\,°C$ stocks, were grown overnight and then used to subculture fresh medium to an initial $OD_{600}$ of 0.004. Cultures were prepared in triplicates for each growth and reduction experiment. $OD_{600}$ and Cr(VI) reduction for each culture were measured at 6, 12, 24, 48, 72, 96, and 120 h. A Cr(VI) reduction assay was performed as described by *Urone (1955)*, with minor modifications. Briefly, one mL of each culture was centrifuged at 7,000 rpm for seven minutes to remove biomass. Ten μL of the supernatant were added to 966 μL of deionized water to which 3.40 μL of sulfuric acid was added. 1,5-Diphenylcarbazide (20 μL) was added to the sample and allowed to set for 10 min for full color development. Readings were taken on a Cary UV-Vis spectrophotometer (Agilent technologies, Santa Clara, CA) at 540 nm. Negative (uninoculated) controls of 50% TSB with 2 mM $K_2CrO_4$ were used to determine TSB-specific abiotic reduction of Cr(VI). In addition, relative Cr(VI) reduction rates were calculated following *Zhu et al. (2008)*.

Resistance determinations were performed in 5 ml of 50% TSB to which $K_2CrO_4$ was added to final concentrations of 5, 10, 20, 40, 60, 80, and 100 mM. Cultures were prepared in triplicate for each resistance experiment and the $OD_{600}$ was taken at 0 and 96 h. Growth was determined against an abiotic control.

## Metabolic screening

All four isolates were screened for their metabolic potential using Biolog's GP2 plates (Hayward, CA) following the manufacturer's protocol with minor modifications. Specifically, isolates were grown on Biolog Universal Growth (BUG) agar (Biolog, Hayward, CA) at 30 °C after which biomass was transferred to 25 ml of inoculating fluid (IF) (Biolog, Hayward, CA) containing sodium thioglycolate to achieve an $OD_{600}$ between 0.68 and 0.75. An aliquot (150 μl) was placed into each well and then the plates were incubated for 22 h at 30 °C. Plates were read using Biolog's OmniLog program (version 1.2.01) which corrects for any difference in $OD_{600}$. Absorbance values were converted to 0(-) for negative, 1(+) for positive, and .05(+/−) for borderline by the Biolog OmniLog program. Replicates with a minimum of two positives or borderlines or one positive and one borderline were considered positive for the metabolite. A total of 95 metabolites were tested per isolate (Table S6). To ease data analysis, metabolites were grouped into 6 guilds (*Zak et al., 1994*): (1) carbohydrates, (2) carboxylic acids, (3) amino acids, (4) amines and amides, (5) polymers and (6) miscellaneous. The total number of positives for each guild was then calculated for each isolate and plotted using principal components analysis (PCA) in PAST3 (*Harper & Ryan, 2001*).

## Sequencing, de novo assembly, and analysis

Whole genome shotgun sequencing was performed by multiplexing the genomic DNA onto one lane using the Illumina HiSeq 2000 platform with 100 bp paired end reads using V2 chemistry at Cincinnati Children's Hospital Medical Center's Genetic Variation and Gene Discovery Core facility. Raw Illumina genomic reads were trimmed of their adapter sequences using the default setting of the program Trimmomatic (version 0.27) (*Lohse et al., 2012*). Trimmed reads were checked for quality using FastQC (version 0.10.2; http://

www.bioinformatics.babraham.ac.uk/projects/fastqc/) and then trimmed for quality using the fastx_ trimmer (-Q33 -l 70) and fastx_ quality_ filter (-Q33 -q 30 -p 50) functions of the FastX toolkit program (version 0.13.2) (http://hannonlab.cshl.edu/fastx_toolkit/). To determine the appropriate range of Kmer length for assembly, the program KmerGenie (version 1.5856) was used (*Chikhi & Medvedev, 2013*; *Zerbino & Birney, 2008*). Cleaned pair end reads were assembled using the default setting of the assembly programs Velvet (version 1.2.10) (*Zerbino & Birney, 2008*) and AbySS (version 1.3.6) (*Simpson et al., 2009*) at a range of Kmer surrounding the estimated Kmer size from KmerGenie. A third assembly was completed using the raw pair-end reads and the intergrated a5 pipeline assembly (*Tritt et al., 2012*). This pipeline automates the processes of data cleaning, error correction, contig assembly, scaffolding, and quality control. The draft assemblies were compared and the best assembly picked for each isolate based on their total contigs, N50, genome size, max contig length, and mean contig length. AbySS assemblies for all four isolates were used for all downstream analyses (Table S3). The average coverage of each genome was >200×, with each assembly containing 30–81 contigs (Table S2). The contigs from the selected assembly were annotated and analyzed using the Departments of Energy's Joint Genome Institutes IMG program (*Markowitz et al., 2012*). Utilizing IMG's data annotations, Pfam categories and their broad category gene counts for the respective genomes were extracted and normalized based on the sum of each row. Principal component analysis (PCA) was then performed using PAST3. All protein coding genes from the annotated draft isolate genomes were submitted to the Pacific Northwest National Laboratory's Species Parallel and Orthology Solver (SPOCS) for analysis to determine pairs of orthologous and paraorthologous proteins between the closely related isolates (*Curtis et al., 2013*).

## Analysis of genomes for the presence of chromate-related genes

To increase the potential for finding chromate related genes, assembled contigs from the four representative isolate genomes were searched against a protein database of chromate related genes acquired from the UniprotKB database (http://www.uniprot.org/). Three databases were downloaded containing the chromate related protein sequences of ChrA (efflux pump), ChrR (reductase), and YieF (reductase). Databases were obtained from text-based queries within the UniprotKB database. Because IMG based annotations did not return any chromate-related genes, assembled contigs were searched manually against individual databases with the BLASTX algorithm using a minimum e-value cutoff of 1e–05. Resultant nucleotide sequences with corresponding hits were translated using the translation tool from ExPASy (http://expasy.org/) and examined to see if the sequences were found within a larger open reading frame (ORF). For each sequence, the top sequence corresponding to an annotated protein with the highest bit and query coverage was selected. The resultant amino acid sequence was searched using the protein BLAST tool from NCBI to further validate the gene candidate based on domain hits as shown within BLAST. Sequences were further scrutinized by examining the top resultant hit within IMG using their alignment tools and gene neighborhood viewer.

Phylogenetic trees of the putative genes were made by using classic *chrR* and *yieF* genes downloaded from UniProtKB as well as from the top hits from homology searches within IMG. The protein sequences were aligned using ClustalW. A phylogenetic tree was then constructed using Maximum likelihood (bootstrap = 50) within the software MEGA (version 6.06) (*Tamura et al., 2007*).

## RESULTS & DISCUSSION

### Bacterial isolation and characterization of chromate resistance and reduction

Four bacterial strains isolated from chromium contaminated soil samples (collected in Seymour, IN) were studied for their ability to resist and reduce Cr(VI). When introduced to Cr(VI), each isolate was able to grow and survive in a minimum of 2 mM chromate but differences in their growth were noted. Overall, three isolates (Cr-K20, Cr-K29, and Cr-K32) grew to relatively dense cultures (OD $\geq$ 4.0) and had similar growth rates whether grown aerobically with or without Cr(VI) (Figs. 1A–1D and Table S1). In contrast, Cr-K1W doubled in optical density when grown without Cr(VI) and it reached exponential growth earlier than the other isolates (Fig. 1A). When experiments were done with Cr(VI), Cr-K1W was also able to maintain a stable stationary phase for over 100 h, which was not observed for the other isolates under similar conditions. Chromate toxicity appeared to have the greatest effect on Cr-K1W when compared to the other three isolates.

The isolates also had different abilities to reduce and resist chromate stress. Cr-K29 and Cr-K32 were capable of reducing 2 mM chromate within 48 h (Figs. 1C and 1D ) and had the relatively fastest reduction rates (Table S1), while resisting up to 100 mM of chromate. Conversely, Cr-K1W and Cr-K20 were only able to resist up to 10 mM and reduced less than 1.2 mM of chromate after 120 h (Figs. 1A and 1B). These two isolates also had the lowest relative Cr(VI) reduction rates (Table S1). Taken together, a connection between reduction and resistance was evident. Chromate resistance has been shown to be connected to the efflux pump gene *chrA*, part of the CHR superfamily, (e.g., *Diaz-Perez et al., 2007*; *Henne et al., 2009a*; *Ramirez-Diaz et al., 2008*), which can provide a wide range of resistance, 0.3 mM–200 mM (*Viti et al., 2013*). However, the link between reduction and resistance in these isolates may point to the importance of a Cr(VI) reduction mechanism and other non-chromate specific responses in aiding resistance, instead of Cr(VI) resistance being driven solely by an efflux pump.

### Metabolic fingerprinting of the isolates

Metabolic fingerprinting found variation between that isolates that is comparable to their growth differences. A survey of 95 potential metabolites showed a core group of 32 metabolites used among all four isolates (Fig. S1). Furthermore, isolates Cr-K1W and Cr-K20 shared the ability to use an additional seven metabolites, while Cr-K29, Cr-K20, and Cr-K32 shared the ability to use 16 additional metabolites (Fig. S1). Of the four isolates, Cr-K20 had the most unique combination of metabolites utilized ($n = 16$). Both Cr-K29 and Cr-K32 exhibited the largest amount of overlap in substrate utilization, which may be related to why these isolates had very similar growth patterns (Figs. 1B and

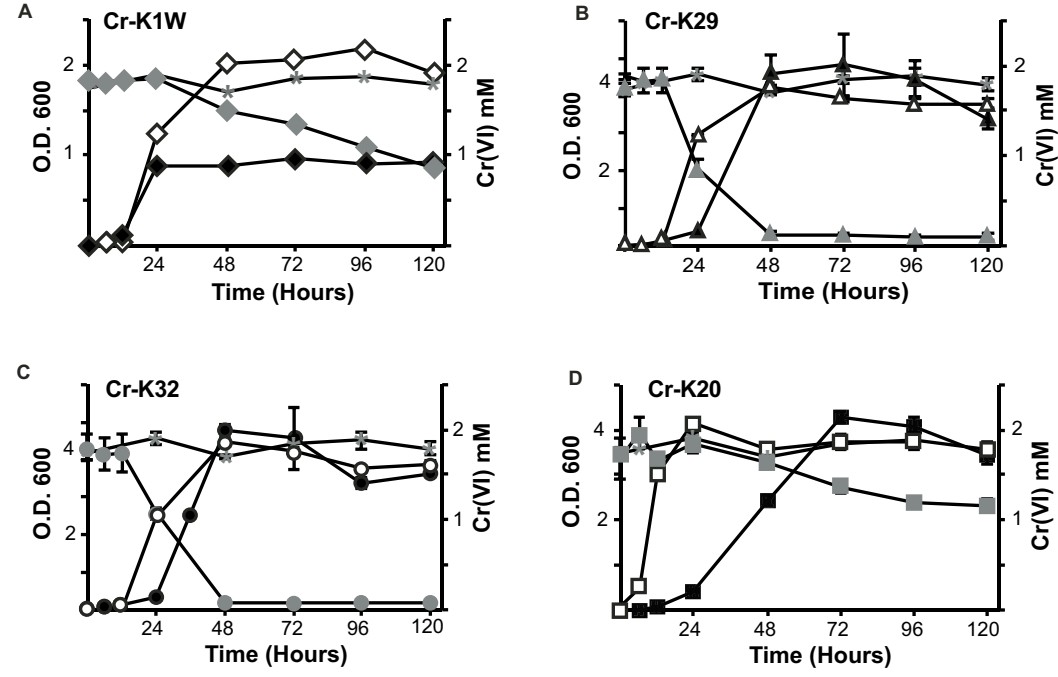

**Figure 1 Comparison of the isolate's ability to grow and reduce chromate.** Chromate reduction (gray fill) and growth curves of *Microbacterium spp.* in the presence (black fill) or absence (no fill) of chromate. Negative (uninoculated) controls for chromate reduction are labeled in each graph with an asterisk. (A) Cr-K1W, diamond; (B) Cr-K29, triangle; (C) Cr-K32, circle; (D) Cr-K20, square ($n = 3$, error bars are SD).

1C). The growth patterns of Cr-K1W and Cr-K20 were unique when compared to the other isolates, which was mirrored by their metabolic fingerprint. Of the six-substrate guilds, Carbohydrates and the Miscellaneous guilds exerted the greatest influence on separating Cr-K1W and Cr-K20 from Cr-K29 and Cr-K32 (Fig. 2). The four remaining substrate guilds (Amines/Amides, Carboxylic acids, Amino Acids, and Polymers) appeared to help separate Cr-K1W away from the other three isolates (Fig. 2).

## Genomic characterization of chromate reducing isolates

The four draft genomes obtained in this study had lengths between 3.79 and 3.91 Mbp with a GC content of ∼68% (Tables S2 and S3). Analysis of the 16S rRNA genes indicated that all four isolates had identical (1,405 bp) 16S rRNA sequences, and that they are *Microbacterium* sp. (*Actinobacteria* phylum), specifically members of the *Microbacterium oxydans* clade (Fig. S2). Gene annotation of the four genomes documented between 3,616 and 3,806 predicted protein coding genes (Table S2). Genomes were assessed to be nearly complete with all genomes containing the 35 universal single copy marker genes (*Raes et al., 2007*) (Table S4).

Analysis of the genomes indicated that the four taxonomically closely-related chromium reducing *Microbacterium* isolates have a surprising amount of inter-strain genomic variation. A broad based genome comparison of the four isolates showed that they share a large predicted protein core made up of 2,810 proteins (Fig. S3). In addition to the

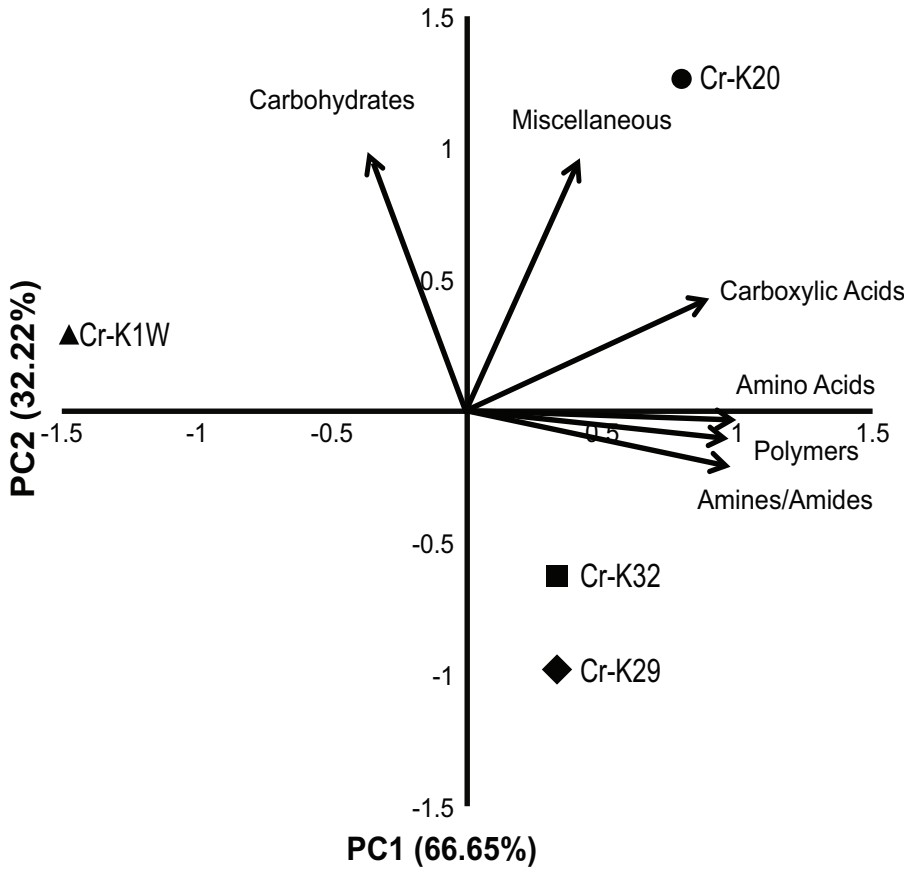

**Figure 2** **PCA of metabolic fingerprinting.** Principal component analysis plot of the four isolates based on their ability to utilize a metabolic substrate.

protein core, isolates Cr-K29 and Cr-K32 shared an additional 602 predicted proteins while isolates Cr-K1W and Cr-K20 shared 854 predicted proteins (Fig. S3). When the annotated genomes were analyzed by Principal Component Analysis, using annotated Pfam proteins (Table S5), a similar grouping pattern was observed (Fig. 3). Of the 21 Pfam categories, Carbohydrate Transport and Metabolism, Transcription, and Inorganic Ion Transport and Metabolism, General Function, and Function Unknown were the most important factors separating the annotated draft genomes (Fig. 2). Genome analysis also confirmed the presence of certain metabolic genes that connect to the metabolic fingerprinting. For example, the genome of all four isolates contained genes needed to utilize sucrose (glycosidases, starch and sucrose metabolism KEGG pathway) and all four isolates were capable of doing so. Further, annotated genes for arabinose (L-arabinose isomerase and L-ribulose 5-phosphate 4-epimerase, pentose and glucuronate interconversions KEGG pathway) and xylose metabolism (xylose isomerase, pentose and glucuronate interconversions KEGG pathway) were only found for Cr-K1W and Cr-K20 and these were the only two isolates capable of utilizing those substrates.

Bacterial chromate stress has been shown to impact broad metabolic functions. For example, a proteomics study of *Arthobacter* FB 24 demonstrated that, when the

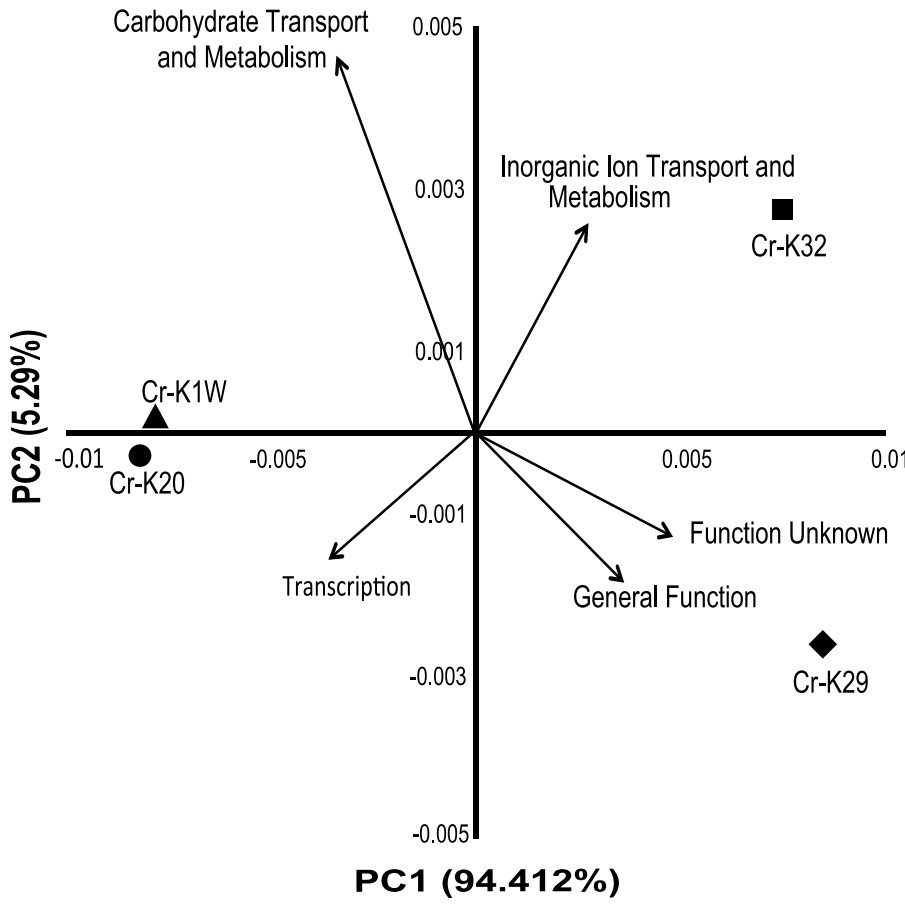

**Figure 3 PCA of Pfam catagories.** Principal component analysis of 21 Pfam catagories from IMG annotations of the four *Microbacterium* sp. genomes. Top five eigenvalues were then plotted as the vectors after analysis.

bacterium was exposed to chromate, a large number of proteins involved in metabolism (e.g., carbohydrate, energy production, and amino acid transport and metabolism) had altered expression (*Henne et al., 2009b*). In *S. oneidensis* MR-1, chromate stress response caused variation in iron and sulfur transport genes and also DNA repair genes (*Brown et al., 2006*). Few genomic differences that related to sulfate transport are evident in the examined *Microbacterium* genomes. The faster reducers, K-29 and K-32, did have two additional genes that are related to iron transport (ABC-type $Fe^{3+}$ transport system, permease component, and siderophore synthetase component). Thus these may be related to their increased ability to thrive in high chromate conditions.

## Genomic annotations related to chromate reduction and resistance

The genomes of the four isolates examined contained putative chromate reductases. An automated IMG annotation of the isolate draft genomes failed to identify chromate reductases or resistant genes. Consequently, custom searches were performed to further examine the genomes for chromate related genes. Manual BLASTX searches were

performed using the UniprotKB databases for ChrA, ChrR, and YieF. The search with known chromate transporters, ChrA homologues, returned no hits within any of the four isolate assembled genomes. While past studies have found that the efflux pump gene, *chrA,* can provide chromate resistance (e.g., *Diaz-Perez et al., 2007*; *Henne et al., 2009a*; *Ramirez-Diaz et al., 2008*), the lack of this gene could suggest that other efflux pumps may be exporting chromate. The genome of each of the isolates did contain two metal associated efflux pumps (arsentite efflux pump and cation/multidrug efflux pump). Alternatively, the physiological link between reduction and resistance could suggest that Cr(VI) reduction is responsible for providing resistance to these isolates.

Curated genome searches were able to identify multiple putative chromate reductases. BLASTX searches against the ChrR UniprotKB database returned a hit in all four isolate genomes. The identified genes were homologues (50–51% identity) to a ChrR reductase in *Thermus scotoductus* (*Opperman, Piater & Van Heerden, 2008*). IMG classified these genes as oxidoreductases and part of the Old Yellow Enzyme family. Analysis of Cr-K32's protein (microk32_01893, 2529442617) using IMG's homolog toolkit, showed a 100% homology to the protein found in Cr-K29 (microk29_00100, 2529452261) and 89% in Cr-K20 (microk20_02158, 2529450446) but only 35% percent identity to the protein found in Cr-K1W. Since the sequence homology of the protein found in Cr-K1W was low, additional searches within this genome were conducted. The genome of Cr-K1W did have a protein identified as an NADH:FMN oxidoreductase from the Old Yellow Enzyme family (cellulok1w_02457, 2529446915) that shared 100% homology to the *chrR*-like gene found in Cr-K20. Comparison of the four closest homologs' gene neighborhoods within IMG showed high similarity of gene composition near the oxidoreductase despite the varying percent identity (Fig. S4). Further, jackhmmer searches (http://www.ebi.ac.uk/Tools/hmmer/search/hmmsearch), of the aligned genes found 51% protein identity, 68% similarity, and with an e-value of 3E–100 to *Thermus scotaductus*' chromate reductase.

A similar curated search with YieF (ChrR sequence homologue) was also able to identify putative chromate reductases. The top result from a BLASTX search using the YieF databases identified a predicted flavoprotein in each of the individual isolates. Alignment of the four individual isolate genes within IMG's homology toolkit revealed high degrees of sequence similarity with Cr-K1W (cellulok1w_02457, 2529446915) and Cr-K20 (microk20_03386, 2529451669) having 100% homology and Cr-K29 (microk29_00694, 2529452855) and Cr-K32 (microk32_00074, 2529440805) having 98% homology to one another. In addition, the genes were also homologues (65% for Cr-K29 and Cr-K32 and 69% for Cr-K1W and Cr-K20) to a predicted chromate reductase (ARUE_c41610) in an *Arthrobacter sp.* (*Niewerth et al., 2012*). Gene neighborhoods of the resultant genes did not show as much conservation compared the putative ChrR-like genes, with only small regions around the gene being shared between the fastest reducers (Cr-K32 and Cr-K29) and slower reducers (Cr-K1W and Cr-K20) but no shared genes between all four (Fig. S5).

Automated annotations of the isolate genomes failed to provide any putative chromate reductases, however, custom database searches were more successful. Each genome did contain genes with sequence homology to the chromate reductases, *chrR* and *yieF,* of

non-model organisms. The putative *chrR*-like genes found in the isolates are homologous to a *chrR* gene (GenBank accession number AM902709) found in *T. scotoductus,* which has been experimentally shown to reduce chromate (*Opperman, Piater & Van Heerden, 2008*). The putative *yieF* genes found in the isolates were homologous to an annotated chromate reductase in *Arthrobacter sp.* RUE61a, a known chromate reducer. Despite being found in all four isolates, the genes did showed sequence variability between the four isolates. Isolates Cr-K29 and Cr-K32 are the faster reducers and their putative genes share more sequence homology when compared to Cr-K1W and Cr-K20, the slower reducers. Phylogenetic trees of both the putative ChrR and YieF genes also displayed this variability with slow reducers or fast reducers being more closely related to one another (Figs. S6 and S7). Neither putative gene was found to group with known chromate reducers, however, this may be an artifact from the lack of known chromate specific genes in *Microbacterium*. While the reasons for the variable ability for each isolate to reduce chromate are still not well understood, the differences observed between the isolates may be related to sequence similarity, expression, or genome content.

## CONCLUSION

Understanding microdiversity is vital for the systematic understanding of how bacterial strains, and populations, can impact biogeochemical processes (*Fuhrman & Campbell, 1998*; *Jaspers & Overmann, 2004*; *Johnson et al., 2006*). Previous studies have documented genomic variation among closely related strains, which some term "ecotypes" (e.g., *Hunt et al., 2008*; *Rocap et al., 2003*; *Welch et al., 2002*). Depending on the bacterium, inter-strain variation can have an impact on microbial function (*Coleman & Chisholm, 2010*; *Martiny, Coleman & Chisholm, 2006*) and at times this connection to function is nonexistent or not as clear (*Meyer & Huber, 2014*; *Simmons et al., 2008*; *Wilmes et al., 2010*). Though a "core" metabolic and genomic structure was seen among the four isolates, our data suggests that Cr(VI) reduction discrepancies within these isolates could be related to strain-level genetic and metabolic variation. Further, chromate resistance may be intertwined with the ability of a bacterium to reduce and transport chromate as well as the type of stress response the organism might have. Fundamentally, the genomic variation between these isolates may point to bacterial adaptation in response to long-term exposure to multiple contaminants (e.g., lead, Cd) including Cr(VI). Further analysis of these discrepancies will help define a Cr(VI) reduction mechanism within these isolates, and will lead to a greater understanding of the importance of inter-strain variation in microbial communities.

## ACKNOWLEDGEMENTS

We would like to thank Dr. C. Titus Brown, Dr. Ian Dworkin, and Dr. Istvan Albert for consultation on computational methods during the MSU Next Generation Sequencing course. The US Environmental Protection Agency, through its Office of Research and Development, collaborated in the research described herein. Any opinions expressed in this paper are those of the authors and do not necessarily reflect the views of the agency; therefore, no official endorsement should be inferred. Any mention of trade names or

commercial products does not constitute endorsement or recommendation for use. This is contribution number 60 of the CMU Institute for Great Lakes Research.

### Funding

This work was supported by the Central Michigan University's College of Science and Technology, Department of Biology, and the Department of Earth and Atmospheric Sciences. The funders had no role in study design, data collection and analysis, decision to publish, or preparation of the manuscript.

### Grant Disclosures

The following grant information was disclosed by the authors:
Central Michigan University's College of Science and Technology.
Department of Biology.
Department of Earth and Atmospheric Sciences.

### Competing Interests

The authors declare there are no competing interests.

### Author Contributions

- Michael W. Henson conceived and designed the experiments, performed the experiments, analyzed the data, wrote the paper, prepared figures and/or tables, reviewed drafts of the paper.
- Jorge W. Santo Domingo conceived and designed the experiments, contributed reagents/materials/analysis tools, wrote the paper, reviewed drafts of the paper.
- Peter S. Kourtev performed the experiments, contributed reagents/materials/analysis tools, wrote the paper, reviewed drafts of the paper.
- Roderick V. Jensen contributed reagents/materials/analysis tools.
- James A. Dunn performed the experiments.
- Deric R. Learman conceived and designed the experiments, analyzed the data, contributed reagents/materials/analysis tools, wrote the paper, prepared figures and/or tables, reviewed drafts of the paper.

### DNA Deposition

The following information was supplied regarding the deposition of DNA sequences:
This BioProject has been deposited at DDBJ/EMBL/GenBank under accession PRJNA236112.

### Supplemental Information

Supplemental information for this article can be found online at http://dx.doi.org/10.7717/peerj.1395#supplemental-information.

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
