# Peer review of "Metabolic and genomic analysis elucidates strain-level variation in Microbacterium spp. isolated from chromate contaminated sediment"

_PeerJ, doi:10.7717/peerj.1395_

## Round 0.1 · original submission · Major Revisions

While addressing the comments of both reviewers, please pay special attention to the major concerns expressed by Reviewer 2.

·

Basic reporting

In general, the manuscript by Henson et al. examining the impact of genomic microdiversity within four highly resistant chromate reducers on metabolic capability and chromate reduction rates is very well written. However, I do suggest that the authors incorporate the comments/revisions listed below to improve the overall expression and readability of the manuscript text.

(1) On p. 5, line 119, "Carry" should be changed to "Cary" UV-Vis spec from Agilent Technologies.
(2) On p. 6, line 125, authors should say either "...added to a final concentration of 5, 10, 20, ..." or "...added to final concentrations of 5, 10, 20, ...."
(3) On p. 7, lines 164-165, I suggest improving the clarity of the sentence by changing to "the average coverage of each genome was >200X, with each assembly containing 30-81 contings" or something similar.
(4) On p. 9, line 195, subheading should be changed to "Results&Discussion."
(5) p. 12, line 256, improve expression by changing to "...all four isolates were capable of doing so" or "all four isolates were able to do so."
(6) p. 12, line 258, "isolate" should be plural.
(7) p. 13, line 285, a semicolon is not correctly used here. The punctuation should be a colon.
(8) p. 14, line 300, "data" is plural, not singular.

Experimental design

Overall, the experimental design is sound and an appropriate level of experimental detail is provided.

Validity of the findings

No comments

Additional comments

I would have liked to have seen more detailed discussion on the genomic differences between the four isolates for genes with annotated functions in the 3 Pfam categories (i.e., Carbohydrate Transport & Metabolism, Transcription, and Inorganic Ion Transport & Metabolism) that were indicated by PCA analysis as the most important factors separating the isolates. For example, were there genes with sequence homology to chromate transporters or iron transporters (e.g., siderophore-mediated iron acquisition systems), which have been shown by others to play an important role in chromate stress responses? Perhaps the inter-strain physiological/functional differences in terms of growth in the presence of chromate and chromate reduction have more to do with genetic differences in chromate stress response pathways than with differences in chromate reductases.

·

Basic reporting

The basic reporting is fair although can be improved along the lines suggested in my general comments to the authors (Section A).

Experimental design

Can be notably improved, as stated in my comments to the authors regarding experimental procedures (Section B, points 2 and 3).

Validity of the findings

I have major concerns in this regard, mainly related on the superficial genome analysis, as detailed in my comments to the authors (Section B, point 4).

Additional comments

Overview:

The manuscript by Henson and colleagues reports on the physiological characterization and genome sequence analysis of 4 closely related Ch(VI)-tolerant Microbacterium sp. isolates recovered from a contaminated site in IN. Based on the analysis of growth curves and time-dependent Cr(VI) reduction in the presence of 2mM Cr(VI) the authors show differences among the strains in Cr(VI) tolerance and reduction capacity. As a matter of fact, one of the key findings the authors claim to have made is the link observed between Cr(VI) reduction capacity of their isolates and with the resistance phenotype.
The authors also determined biolog-based of metabolic profiles of the 4 strains to extend the phenotypic/physiologic characterization. Finally, draft genome sequences were generated for each isolate and automatically annotated at the IMG server and the resulting annotations refined for chromate reduction and resistance determinants by means of blast-based searching of the genomes against members of two well characterized chromate reductase families (ChrR from Pseudomonas putida and YieF from E. coli) and the ChrA efflux pump. Only the latter approach succeeded in identifying ChrR and YieF homologues in the query genomes.

A. General Comments.
The manuscript has a good introduction on the impact of Cr as a major environmental pollutant and its bio-remediation strategies using microbes. However, not much background information is provided on the mining of bacterial, and specifically Microbacterium genomes, regarding chromate reduction and resistance determinants. The genes and mechanisms involved, relevant to the genome analysis performed by the authors, should be explicitly stated in the introduction instead of burying them in reference lists. In other words, an effort needs to be made to better relate key background information with the relevant findings of their study. This important information should not be buried in long reference lists.
The manuscript is well and clearly written, although it may be a bit too verbose. The quality of the artwork can certainly be improved and associated legends are not always clear or explicit enough for understanding without reference to the main text, as will be detailed below. My major concerns are related to the superficial genome analyses performed.

B. Major comments
1. Lines (L10-13) of the abstract are vague and even misleading. I don't think that the authors have performed a detailed analysis of the (genomic) differences displayed by the 4 strains. They don't state the main specific findings of the study: link observed between Cr(VI) reduction capacity of their isolates and with the resistance phenotype; presence of ChrR and YieF homologues in the genome sequences …). What about efflux pumps?

2. Section on Genome Annotation (starting at L175).
I think that the title of the section is misleading, as the focus is only on detection homolougues of the ChrR and YieF chromate reductases. The procedure is not clear and even cumbersome. What strategy did the authors use to generate the ChrA, ChrR and YieF specific databases? Did they use text-based queries or homology-based (i.e. blast-based) queries of the the UniprotKB DB? Why did the authors use blastx of the contigs against the custom Dbs, instead of using the translation products of the CDSs identified by the HMMs (gene prediction software) run by the IMG pipeline for direct blastp searching against the ChrA, ChrR and YieF specific databases? What query coverage did the authors impose in their blast searches. Filtering by e-value (1-e-5) is highly prone to selecting spurious hits, that have only local (domain-specific) homology.
Given the large genetic distance between the well characterized ChrA ChrR and YieF reference proteins (mainly from Gammaproteobacteria) to the query genomes (Actinobacteria), a much more sensitive approach would have been to run hmmer3 searches using family-specific HMMs.

3. Section on Genomic characterization of the isolates (starting at L236)
The authors state in the L157-160 that they used 3 programs (AbySS, Velvet and a5) to assemble their genomes and selected the best one for downstream analyses. But the authors don't state which assembler produced the best assemblies. A comparative table in supplementary materials could clarify this.

4. Results section starting at L260 (Genomic annotation of Cr resistance and reduction determinants)
The tables 1 and 2 are really of little use. I would repleace them by a proper phylogenetic analysis of bona-fide homologues of the ChrR and YieF protein families, adequately describing the global level of identity and query coverage. Are there multiple copies of ChrR and YieF family members in the genomes under study? Are they recovered in monophyletic clusters containing characterized chromate reductases? What about the genomic context of the putative ChrR and YieF homologues in your strains? Is there evidence of microsynteny around the target genes that further supports the inforence of homology? Some comparative genome maps of these key regions would be very useful. Is there evidence for horizontal movement of these reductases? It is striking that the putative Cr-K1W homologue displays only 35% identity with the other Microbacterium proteins. And what about such homologues in other Microbacterium genomes? Again, comprehensive, well-designed and performed phylogenetic analysis could illuminate all these questions.
And what about ChrA homologues? No mention of their presence in the target strains is made. It is certainly the interplay between efflux pumps and Cr reductases what confers the observed phenotype. By the way, there are 6 large efflux pump superfamilies (RND, MATE, MFS, ABC …). To what superfamily does ChrA belong? Many have a very broad substrate specificity. I don't think the authors can claim that only ChrA is involved in Cr efflux (L214). This has to be studied in much greater detail. What is the repertoire of efflux pumps in your strains?
The discussion on all these issues needs to be improved and deepened. The authors just scratch on the very surface of this complex issue. The result is an almost trivial discussion lacking adequate depth and rigor, based on a very superficial genome analysis.

B. Minor comments.
L54-55. This statement sounds dangerously Lamarckian in my ears … The authors may re-formulate the statement

L91. Define TSB here, not in L99.
L113. and growth rates should read of growth?
L114 Urone not italics
L117 adjusted to X pH
L147. I assume that the 4 genomes were multiplexed and sequenced on a single MiSeq cell. Please specify this point and state the chemistry version used for the sequencing.
L197-207. The description is excessively wordy. You may include the strain names on top of each panel in figure 1 to make it clearer. Labeling of the right axes seems not correct. I understood that only one concentration (2mM) was used in the experiments. If so, completely remove that axis and properly state in the figure caption.
L224-225: strain Cr-K20 in mentioned in both lines. Is this correct?
L282. Query identity? The identity is always defined for pairwise comparisons. It is completely unclear if you are meaning local, blast-hit identity or global identities from global alignments. In general, what query coverage did the authors impose in their blast searches. Filtering by e-value (1-e-5) is highly prone to selecting spurious hits, that have only local (domain-specific) homology, as commented previously.
L168,493 and 496. PCA is Principal Component Analysis.

---

## Round 0.2 · accepted · Accept

Thank you for addressing the concerns of both reviewers.

·

Basic reporting

No comments.

Experimental design

No comments.

Validity of the findings

No comments.

Additional comments

The authors, in my opinion, have adequately responded to comments and suggestions and have made appropriate revisions to the manuscript. My decision is that the manuscript is ready for acceptance.